# Northern Pulmonary Hypertension: A Forgotten Kind of Pulmonary Circulation Pathology

**DOI:** 10.3390/life14070875

**Published:** 2024-07-13

**Authors:** Djuro Kosanovic, Sergey N. Avdeev, Andrey P. Milovanov, Andrey L. Chernyaev

**Affiliations:** 1Department of Pulmonology, I. M. Sechenov First Moscow State Medical University (Sechenov University), 119991 Moscow, Russia; serg_avdeev@list.ru; 2Laboratory of Pathology of Reproduction, A. P. Avtsyn Research Institute of Human Morphology of FSBSI “Petrovsky National Research Centre of Surgery”, 117418 Moscow, Russia; a_p_milovanov@mail.ru; 3Fundamental Pulmonology Department, Federal Pulmonology Research Institute, 115682 Moscow, Russia; 4Laboratory of Clinical Morphology, A. P. Avtsyn Research Institute of Human Morphology of FSBSI “Petrovsky National Research Centre of Surgery”, 117418 Moscow, Russia

**Keywords:** pulmonary hypertension, pulmonary circulation, cold environments, Russian North, polar regions, pulmonary vasculature

## Abstract

Northern pulmonary hypertension (NPH) is a medical condition that is still enigmatic in non-Russian-speaking countries. The extant previous literature is mostly available in the Russian language and, therefore, not accessible to the rest of the world. The recent increased interest in climate changes and environmental effects on pulmonary circulation prompted us to summarize the knowledge from the past about the effects of cold on pulmonary vasculature. In this review, we, for the first time, describe, in detail, the pathological attributes of human NPH, a medical disorder that occurs in people living in extremely cold regions, in the English language. Briefly, NPH is characterized by the hyper-muscularization of the pulmonary arteries and de novo muscularization of the arterioles with the ultimate development of right ventricular hypertrophy. However, the profound molecular mechanisms of the NPH pathology are to be revealed in future comprehensive studies.

## 1. Introduction

In the period from 1968 to 1987, the USSR was actively involved in the development of the circumpolar and polar regions of the European and Asian North of the country. The researchers had a difficult task in identifying changes in the organs and tissues of a person who came to develop these regions characterized by the extreme climate (Figure 1). At the same time, solving the problems of geographical pathology of the cardiovascular system in the visiting Northerners and the deciphering of climate–geographical relations explaining the possible unfavorable effects of extreme environmental factors on the cardiac system, respiratory system and peripheral blood circulation was of great importance [1]. When analyzing the literature devoted to bioclimatic relationships, the external factors with real influence on human blood circulation were singled out. Among these factors, the most important one regarding the degree of relevant effect is sharp seasonal, inter- and intraday atmospheric pressure variations, the so-called barometric pits, especially characteristic of the Kola Peninsula (European North), the extreme northeast (Asian North) and the entire Polar region of the country [2]. In this review, we have analyzed the knowledge that exists almost entirely in the Russian language only, and therefore, it is not available to the rest of the world. Therefore, we have summarized all literature in the Russian language from the past, with a general focus on the effects of extremely cold environments on the lung tissue and its circulation.

## 2. Environmental Characteristics of the Cold Regions

The complex environmental conditions characteristic of these regions (Arctic and Subarctic), described in the work of Veselukhin [3], have an important influence on the changes in angiologic and cardiologic indices. These environmental conditions include the following:-Negative deviation of air temperature from the latitudinal mean by 30 °C in January and 10 °C in July;-Low annual sum of effective temperatures with a short frost-free period;-Higher winter severity scores (3.5 Bodman’s units) and its long duration;-Absence (up to 160 days) or lack of solar radiation (from November to February, the radiation balance is negative above the Arctic Circle);-Excessive insolation in summer (on a polar day);-High levels of the snow cover and albedo;-Permafrost soils with abnormal content of mobile forms of macro- and microelements;-Sharp restructuring of the baric field by seasons (in the warm half of the year, the pressure over the land is lowered and that over the ocean is increased, and vice versa in the cold half of the year);-Peculiarity of magnetic and gravitational regimes;-Sparse human population and frequent isolation of micro-collectives; -Remoteness of most areas and irregularity of their communication with the “mainland”.

An important achievement of Northern cardiology was the discovery of morpho-functional reactions of various parts of the heart, pulmonary vessels and peripheral blood vessels, arising in humans during the process of adaptation to the north.

## 3. The Discovery of Northern Pulmonary Hypertension

In the several monographs [4,5,6,7], the cardiovascular system adaptation issues of Northerners–new settlers are covered, and all authors emphasize the variability of general trends of adaptation. Therefore, the disputes continue regarding the hypo- or hypertensive effect of extreme factors of the north, as well as the significance of the T wave in electrocardiography (ECG) analysis, which characterizes peripheral vascular resistance.

Considering the functional state of peripheral blood circulation in Northerners in close relationship with changes in the respiratory system and pulmonary circulation, Milovanov et al. [8] put forward a hypothesis of discordant interaction between pulmonary and peripheral blood circulation, which explains the appearance of hypo- and hypertensive reaction in the same individual depending on the stage of adaptation. To confirm this hypothesis, 252 healthy men aged 20 to 45 years with different polar experiences were examined during two expeditions to Pevek (Asian North) and Naryan-Mar (European North). The control group consisted of the residents of Blagoveshchensk of the same age. During these expeditions, a synchronous study of the blood pressure levels in the systemic and pulmonary circulation was carried out. In the newcomers who had lived in the Asian and European North for less than 12 months, the average value of systolic pulmonary artery pressure (sPAP) in Pevek was 60.4 mmHg, and this was 45.5 mmHg in Naryan-Mar, which was 2.5–3 times higher than the level of sPAP in Blagoveshchensk residents. At the same time, arterial pressure in the brachial artery amounted to 132.1/84.1 and 118.2/79.2 mmHg, respectively.

For the first time, Northern pulmonary hypertension in young, practically healthy residents of the northeast region of Russia was found in Magadan in 1972 [9]. The presence of such pathology was indicated by hypertrophy of the right heart sections in young people who died violent deaths without any other pathology. Further study of the systolic pressure in the pulmonary artery in the visiting, practically healthy population of the Magadan region was performed [10]. It was noted that such hypertension had a benign character and was practically asymptomatic. The presence of such hypertension has not been previously reported either in the domestic or foreign literature. In 1978, Schaefer et al. [11] reported dilatation of the pulmonary artery trunk in Eskimos on radiologic examination, which indirectly confirmed the presence of pulmonary hypertension. Subsequently, an increase in systolic pressure in the pulmonary artery was shown in builders of the Baikal-Amur Mainline and Northern Transbaikalia [12]. Bligh and Chauca [13] observed an increase in pulmonary artery pressure in humans and animals in Arctic conditions. It was shown that cold exposure of sheep and bulls causes a similar effect as hypoxic hypoxia. It was assumed that the interaction of these factors (cold and hypoxia) leads to an increase in pulmonary artery pressure.

In the period from 1972 to 1978, 665 practically healthy residents of the Arkhangelsk and Magadan regions and Yakutia aged 17–52 years, including natives (Nenets, Chukchi, Yakuts), were examined. The measurement of sPAP was carried out using Burstin’s method (external graphic recordings) with modifications by Dembo and Shapkayts [14]. The Burstin method was based on measuring the isovolumetric right ventricular relaxation time by means of the phonocardiogram and jugular phlebogram. The accuracy of this method of indirect determination of sPAP was confirmed by comparisons with direct methods and, therefore, found application in clinical practice [15,16]. The comparison group was 176 practically healthy persons from Moscow and Blagoveshchensk of the same age.

The results of these studies allowed us to introduce a new concept—Northern pulmonary hypertension (Figure 2) [1,10]. In general, sPAP from 23 to 30 mmHg was considered normal. Among the immigrant population of Magadan Oblast, sPAP ranged from 12 to 63.1 mmHg (mean value 33.4 mmHg) and was increased in 58.9% of the examined subjects. In the comparison group, this parameter was 21.2 mmHg, and that above 30 mmHg was found in only 10.8% of subjects. It was shown that the increase in sPAP in males and females was comparable. However, the potential gender differences with regard to frequency or disease severity after cold exposure were not profoundly investigated, and future studies should address this important issue. In the natives, the variation of sPAP was from 19 to 64 mmHg. In the natives of the Magadan region, an increase in pressure above 30 mmHg was observed in 25–80% of observations, depending on the area of habitat. In the Arkhangelsk region (European North), the increase in sPAP was found at 15 to 56%. Interestingly, it was shown that the most pronounced increase in sPAP was found in the inhabitants of the Polar region, especially in those who came from temperate latitudes.

Furthermore, it was revealed that cardiac minute volume and cardiac index in Northerners tend to decrease. Energy consumption of the right ventricle to promote 1 L of blood in the visiting Northerners increased in 70% of the examined, while in the natives, it was observed only in 45%. Further, it was shown that with the increasing length of stay in the north, sPAP also increased in 68.2% of those who lived in these regions for an average of 9.7 years. In those who lived in the north for an average of 13 years, this parameter decreased to 54.5%, which is explained by the fact that those who have adapted to extreme environmental factors remain in the north, and those who have poorly adapted leave the region after 3–7 years. Importantly, Milovanov [1] and Milovanov et al. [8] described certain stages in the formation of Northern pulmonary hypertension. Four stages of the adaptive process in the lungs were emphasized (Figure 3): 

Stage 1 represents the initial period of adaptive pressure (length of residence in the north is 3–12 months). Ventilation/perfusion ratios are unstable and fluctuate significantly in different lung zones, while sPAP increases significantly.

Stage 2 is the period of lung function stabilization (length of residence in the north is 1–3 years). The level of sPAP may decrease slightly compared to stage 1, but the incidence of Northern pulmonary hypertension increases.

Stage 3 is the period of lung adaptation (length of residence in the north is from 3 to 10 years). In this stage, Northern pulmonary hypertension is detected in 80% of observations, and there is a good conjugation of increased blood flow with hyperventilation in the upper and middle parts of the lungs.

Stage 4 represents the period of dysadaptive manifestations (length of residence in the north is 10–15 years) and is characterized by an increase in the values of sPAP and an increase in pulmonary vascular resistance.

## 4. Morphological Alterations of the Respiratory System and Pulmonary Circulation Due to Long-Term Living in Extremely Cold Environments

Northern pulmonary hypertension has quite definite morphological attributes (Figure 4) in the form of moderate myocardial hypertrophy of the right ventricle and characteristic remodeling of vessels of the pulmonary circulation with the phenomena of hypertrophy and disorganization of elastic and reticular fibers, as well as the muscularization of arterioles with predominant localization in the lower parts of the lungs. In addition, the changes occur in the bronchial tree at all levels of branching. According to the data of Milovanov [1], in the newcomers of Pevek who were in the stage of adaptation, sPAP in the upper sections of the lungs averaged 23.5 mmHg, in the middle sections 40.1 mmHg and in the lower sections 48.5 mmHg. Thus, the functional studies of pulmonary circulation, carried out in expedition conditions in different groups of the Northern population, allowed us to analyze the main pathogenic cause of right heart hypertrophy in the Northerners and pulmonary vascular remodeling. Alveoli and the respiratory membrane—the locations of direct gas exchange—have their own peculiarities in the Northern newcomers. Alveoli are often deformed, and interalveolar septa are thickened due to dilated and full-blooded capillaries (pre- and post-capillaries). There is a tendency for hypertrophy of elastic and reticular fibers. Dystelectasis was observed predominantly in the lower parts quite often. Importantly, the morphometry revealed a significant increase in the alveolar surface area by 16% in the Northerners, as compared to the people living in Moscow. It was assumed that the increase in the alveolar surface area is associated with hypertrophy and additional opening of alveoli, which contributes to an increase in the number of functioning capillaries of the interalveolar septa.

For a greater understanding of the processes of the human lung’s adaptation to the conditions of the Far Asian North, the evaluation of the phenomenon of increased capillary bed capacity of alveoli is important [1,9,10,17], and such unusual full-bloodedness was called “progressive proliferative capillaropathy”. Marachev [18] showed that the area of the capillary surface in newcomers of Magadan was increased on average by 20%, and the volume of pulmonary capillaries by 68%. According to Milovanov [1], the increased vascularization of alveoli in Northerners, and, hence, the increased volume in alveolar capillaries, is associated with increased pressure in the pulmonary artery, i.e., Northern pulmonary hypertension. It is suggested that the increase in intra-alveolar pressure during cold exposure of the lungs may be important in the development of such pulmonary hypertension. The “adaptation meaning” of the increase in the capacity of pulmonary capillaries consists in the increase in blood volume directly in the zone of gas diffusion, i.e., in the increase in the blood component of the diffusion capacity of lungs in Northerners. The obtained data indicate that in the visiting Northerners and native inhabitants of the North there is an increase in the length of the “functional zones” on account of the components of the respiratory membrane. The ultrastructural remodeling of the respiratory membrane occurs to the maximum extent of the most dense and compact layer of capillaries of interalveolar septa, as evidenced by the fusion of basal membranes of endothelium and alveolar epithelial type I cells (Figure 4). Such changes indicate higher rates of gas diffusion in the lungs of the indigenous inhabitants of the North compared to the newcomers due to more advanced ultrastructural adaptation of the respiratory membrane. The feasibility of such changes in the conditions of the North is confirmed by the fact that the highest degree of “thinning” of the air–blood barrier components was noted in the reindeer, an animal known to be best adapted to extremely cold environments [19]. 

According to Milovanov [1], the adaptation of pulmonary circulation occurs in the following sequence: enlargement of alveolar capillaries with an increase in the functional zone of the respiratory membrane leads to hypervolemic rearrangement of resistance vessels with subsequent hypermuscularization of large branches of the pulmonary artery and increase in the capacity of distributing arteries, which ultimately contributes to hypertrophy of the right ventricle of the heart. The changes described above are characteristic of adaptation and stress [20].

Disorganization of the elastic framework of the trunk and pulmonary arteries with the development of elastolysis, widening of gaps by elastic membranes with their partial fragmentation and disorderly twisting, straightening of elastic membranes, formation of a muscular layer in the upper third of the media, multiplication of the internal elastic membrane as a component of the intima and formation of circular smooth muscles in the middle third of the media sometimes with hyperelastosis of the intima were revealed in the visiting residents of the Far Asian North. In addition, the appearance of connective tissue “cases” around arterioles and the strengthening of the elastic framework of small veins and venules were revealed. At the level of capillaries of interalveolar septa, ultrastructural study revealed the swelling of capillaries in the lumen of alveoli and thinning of their walls, which increased the area of contact “air-blood” [1]. Northern pulmonary hypertension often has its micro-symptomatology in the form of gratuitous fainting states, dyspnea when performing light physical work, which appears more often in the cold season, palpitations and heart pain, rapid fatigue and general weakness. X-ray examination revealed an increase in the diameter of the pulmonary artery trunk and hypertrophy of the right ventricle of the heart. A second sound accent is audible above the pulmonary artery with great consistency. In persons with sPAP above 40 mmHg, this symptomatology is more clearly manifested. The analysis of the oxygen regime of the organism showed that oxygen tension at all stages of its transport in people with Northern pulmonary hypertension remains normal, but the volumetric rate of O_2_ transport changes significantly, increasing up to 50%, while the indicators of efficiency and economy decrease on average by 30%. Indices of forced pulmonary ventilation consistently indicated increased bronchial resistance. Alveolar ventilation under conditions of relative rest increased, but the amount of O_2_ in arterial and venous blood was markedly decreased due to ventilation-perfusion disturbances in the lungs, while the arteriovenous O_2_ difference was increased because the cardiovascular system and blood compensated ventilation disturbances well. At the same time, the values of maximal and marginal alveolar ventilation rates were halved in those with high sPAP. The values of residual volume, anatomical and functional space and expiratory lung closure significantly exceeded normal values. Lung “hypertrophy” in Northern pulmonary hypertension was constantly accompanied by regional redistribution of alveolar ventilation and blood flow with an increase in the load on the upper and middle parts of the lungs. The obtained results allowed us to draw an important conclusion that the pressure increase in the pulmonary circulation and ventilation disorders are not only interconnected but mutually conditioned, i.e., “polar dyspnea”, which is quite common in newcomers of the north, is closely related to Northern pulmonary hypertension. It was shown that the basis of Northern pulmonary hypertension is the adaptive restructuring of respiratory organs in the harsh climatic conditions of the north, where the leading extreme environmental factor is cold. The increased oxygen demand of the organism is realized by the increased intensification of the functions of respiratory organs, cardiovascular system and blood. Increased tone of muscular arteries and arterioles in response to developing regional alveolar hypoxia and arterial hypoxemia, as a rule, is observed in the lower parts of the lungs. Norepinephrine plays a significant role in increasing the tone of muscular arteries, as the production of catecholamines is increased in Northerners. The same mechanism of regulation of the optimal ratio of ventilation and blood flow in the respiratory parts of the lungs is also observed in some pathologies, like chronic obstructive pulmonary disease and interstitial lung diseases. Consequently, moderate Northern pulmonary hypertension in Northerners can be considered as a necessary stage of regulation of gas exchange or protection at a decrease in the morphofunctional reserve of lungs, passing at the development of pathological process to its opposite and aggravating the course of bronchopulmonary pathology. The fixation of long-term adaptive reactions of the pulmonary circulation in newcomers and native inhabitants of the north is carried out by means of structural reorganization of blood vessels and right heart departments. During prolonged stays in the north, in practically healthy people, the volume of expiratory lung closure increases as a result of decreased elastic traction and dynamic compression of small airways (terminal and respiratory bronchioles). These changes contribute to a decrease in the volume of blood flow in interalveolar capillaries, thereby increasing the hypertensive effect. It is reasonable to distinguish two forms of Northern pulmonary hypertension-adaptive and dysadaptive with the development of latent respiratory insufficiency, which is more common in people aged 35–50 years after 10 years of living in the north. Thus, Northern pulmonary hypertension is widespread among the immigrant and indigenous populations of circumpolar and polar regions. The cause of such pulmonary hypertension is an adaptive morpho-functional remodeling of respiratory organs in response to the extreme impact of climate–geographic environmental factors. According to pathogenic mechanisms and structural changes of pulmonary vessels, Northern pulmonary hypertension should be attributed to the precapillary–capillary type.

## 5. Right Ventricular Hypertrophy in People Living in the Extremely Cold Environments

In order to confirm the causal relationship of right ventricular hypertrophy with the general cooling of the organism, an experimental study on mongrel white rats was performed. White mongrel male rats with an initial weight of 190 g were cooled daily for 6 h in individual cages in a climatic chamber “Foetron” for 6 weeks with a temperature decrease from 5 °C in the first week to −20 °C in the sixth week with its gradual decrease by 5 degrees every week. At the end of each week, animals were sacrificed under anesthesia. After the sacrifice, the hearts of rats were taken out and dissected. The right ventricle was separated from the left ventricle and septum and weighted on analytical scales. The measurement of right ventricular hypertrophy was performed via the calculation of the ventricular index. The ventricular index was calculated as the ratio of the mass of the right ventricle with the corresponding part of the interventricular septum to the same mass of the left ventricle. By the end of the first week of cooling, the rats showed an increase in the inner surface of the right ventricle, a significant increase in the volume density of cardiomyocytes and their diameter against the background of a decrease in capillary density (Figure 5). The parameters of the left ventricle did not change. In the next two weeks of cooling, the animals continued the increase in the right ventricular volume and cardiomyocyte diameter with a significant increase in the diameter of nuclei compared to the control group. Such changes in the morphometric parameters of the rat heart are similar to the changes in the parameters of the heart of the Northerners in the phase of stabilization of their organism. At the end of the fourth week, the rats showed a significant increase in the mass of the right ventricle compared to the control group, while there was a significant increase in the density of cardiomyocytes, their diameter and size of nuclei. On the sixth week of the experiment, the maximum density and diameter of cardiomyocytes were registered against the background of restored capillarization of the right ventricle of the heart. These indicators, in our opinion, testify to the adaptation of animals even at cooling to −20 °C [21].

According to Pivovarov et al. [22], important information was obtained about the adaptation process in Northerners in polar conditions. Basically, in newcomers who lived in the north for less than 3 years, a tendency to decrease the blood volume flowing to the right ventricle and increase the volume flowing away from it was observed. This phenomenon is probably related to the increase in systolic ejection by the right ventricle during this period of adaptation. In the left ventricle during this period, a decrease in the reserve volume was observed, which was comparable to the value of the residual volume of the left ventricular cavity, which corresponds to the vasospastic cold reaction and also possibly to blood deposition by the mechanism of the “Parin reflex” [20,23]. In other words, in the stage of adaptive pressure, there are “functional scissors” between the pulmonary and systemic blood circulation. In people with Northern experience from 4 to 15 years, there is equalization of ventricular volume parameters of the heart, but the value of their reserve volume remains lower than in the comparison group of Moscow people. Despite the obvious minimization of ventricular reserve volume, the established ratio of cardiac volume parameters in Northerners should be reasonable and adequate for the severe conditions of the Magadan region.

In Northerners with polar experience of more than 15 years, the value of cardiac volume parameters was the smallest compared to other groups of Northerners and people living in Moscow as the comparison group. Most likely, this correlation reflects the signs of beginning the process of dysadaptation. The conducted correlation analysis between ventricular inflow and outflow volume parameters showed that the greatest number and strength of connections were observed between 4 and 9 years of polar experience, which once again testifies to the achievement of adaptation in the Northerners during this period of adaptation [24]. Thus, the adaptation of the cardiovascular system in newcomers to the north is a long, complex process of interaction of the pulmonary and systemic blood circulation. In the first years of residence in the north, the right heart area is subjected to maximum stress, which adapts to the extreme effects of the environment due to increased muscle mass. The forming hypertrophy of the right ventricle is a structural representation of a new level of pulmonary hemodynamic-adaptive pulmonary hypertension.

## 6. Effects of Cold Environment on Pulmonary Circulation: Interaction with High-Altitude Hypoxia and Potential Molecular Mechanisms

Importantly, in recent years, we have continued this research direction, providing some new findings with regard to the short- and long-term effects of cold environments on pulmonary circulation [25,26,27]. It is also worth mentioning that the existing knowledge from the rest of the world (outside the Russian-speaking milieu) with regard to this topic is very limited, with a low number of relevant studies in the last 45 years [28,29,30,31,32,33,34]. Early studies in cattle suggested the contribution of cold to high-altitude hypoxia in affecting the pulmonary circulation of those animals [28,29]. Recently, we have investigated the short- and long-term effects of cold exposure in Kyrgyz highlanders [25,26,27]. With regard to the acute effects of cold, the data from our study demonstrated that short-term cold exposure resulted in an increase in pulmonary artery pressure in Kyrgyz high-altitude dwellers [25]. In the following study, we have found an increase in pulmonary artery pressure (estimated by the tricuspid regurgitant systolic pressure gradient), although not statistically significant, in Kyrgyz lowlanders during the cold season [26]. Importantly, the effects of cold were even more prominent in Kyrgyz highlanders and the data showed a significant elevation of pulmonary artery pressure during the cold season, as compared to the warm season [26]. Therefore, the existing results clearly indicated the existence of the interaction between the cold and high-altitude hypoxic milieu.

Although the precise molecular mechanisms underlying the pathology of cold-induced pulmonary hypertension in humans are not known, there are some important data derived from the experimental rat model [33,34]. Two studies revealed that in the rats exposed to cold, there is an upregulation in several signals, including the augmented expression of phosphodiesterase-1C, interleukin-6 and tumor necrosis factor-alpha, as well as increased macrophage infiltration, superoxide production and nicotinamide adenine dinucleotide phosphate oxidase activity [33,34]. Interestingly, these molecular signals and events associated with altered phosphodiesterase, abnormal inflammation and oxidative stress are already well-known as relevant players in the pathology of other forms of pulmonary hypertension. These data are useful to provide a future research direction in the context of human disease.

## 7. Further Implications: Lifestyle, Clinical Practices, Public and Global Health Issues

The cold, as an important ecological factor, is present not only in the geographic locations that are the main focus of this article (north, northeast and Polar regions) but also in high-altitude environments and during the winter time in moderate climates. Therefore, the effects of cold exposure on pulmonary vasculature are of general and global interest. As already mentioned, the human populations living in these environments are often isolated from each other and are significantly remote from the “mainland” [3]. The human settlements of the north and Polar regions are not characterized by the existence of fully contemporary civilization facilities, including good roads, stable electric supply or internet connections. Importantly, these locations possess limited medical resources and equipment and do not have modern clinics. Therefore, medical care and clinical practice are very inaccessible, as indicated by these relevant public health issues. However, the people living in extremely cold environments, particularly the native inhabitants, developed several behavioral and cultural “adaptations” and established a unique lifestyle with regard to the proper clothing against the dissipation of body heat, warm “shelters” and a special diet [35].

Furthermore, the literature suggested that the knowledge about the impact of the cold on pulmonary circulation is of global interest. For example, a study from Israel demonstrated that the values of pulmonary artery pressure were higher during the colder seasons in ambulatory patients who were assessed by echocardiography [36]. Finally, it was shown that there were low-temperature effects on the pathogenesis of other forms of pulmonary hypertension not associated with extreme environments [37]. Briefly, the cold exposure worsens the development of the disease in the experimental monocrotaline model of pulmonary arterial hypertension.

## 8. Limitations

We would like to mention that there are certain limitations of our review article, which should be kept in mind while interpreting the data from the existing literature. First, many relevant studies described in this paper were performed a couple of decades ago, and some of the methods, for example, the assessment of pulmonary hemodynamics, were performed using the old technical approaches. Therefore, new studies using contemporary and more accurate techniques should be applied in the future to validate and prove the old findings. In addition, due to the fact that many literature sources were published several decades ago and written only in the Russian language, it is not easy for readers from other countries to assess the original papers and to discern the accuracy and rigor of the experiments. However, we hope that our paper will intrigue and inspire the researchers in the field to confirm the old findings and provide new data with regard to this important medical phenomenon, such as cold-induced pulmonary hypertension.

## 9. Conclusions

The morphological basis for the “adaptation fee” of the pulmonary circulation in the conditions of the Far North is the inclusion of morpho-functional mechanisms that provide long-term compensation for altered hemodynamics. The adaptation of pulmonary circulation in the visiting Northerners appeared due to “abduction of reserves” of compensation of the pulmonary circulation, which are (1) hypertrophy of the right ventricle of the heart, (2) increase in the capacity of distributing arteries, (3) hyper-muscularization of the walls of large branches of the pulmonary artery and de novo muscularization of the walls of arterioles with simultaneous expansion of a significant number of intralobular branches of the pulmonary artery, (4) coarsening of the elastic framework of veins and venules and (5) the appearance of connective tissue “couplings” around the small branches of the pulmonary artery. Ultimately, this is expressed in the decrease in the general reserve capabilities of the organism with cases of pulmonary pathology development, which imposes increased demands on the same systems.

To date, there are no data in the available literature to indicate whether the alterations of the pulmonary vasculature or right ventricle are reversible or not upon the return of the individuals to an environment not characterized by the extreme cold. Therefore, future studies should focus on resolving this important scientific and clinical issue.

Finally, new data and further investigations of this interesting scientific and medical phenomenon are needed to reveal the underlying molecular mechanisms of the Northern pulmonary hypertension pathogenesis. 

## Figures and Tables

**Figure 1 life-14-00875-f001:**
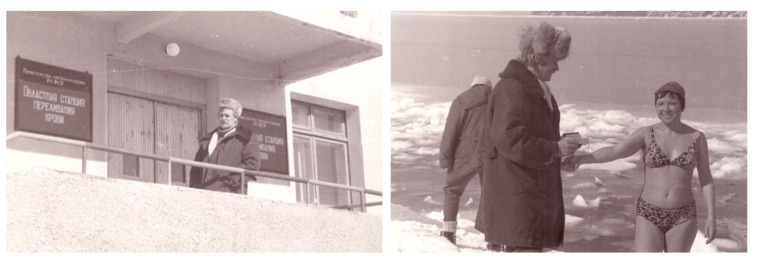
**Scientific expedition to the northeast Russian city of Magadan.** The photographs of the members of scientific expedition to the city of Magadan in 1981 are shown.

**Figure 2 life-14-00875-f002:**
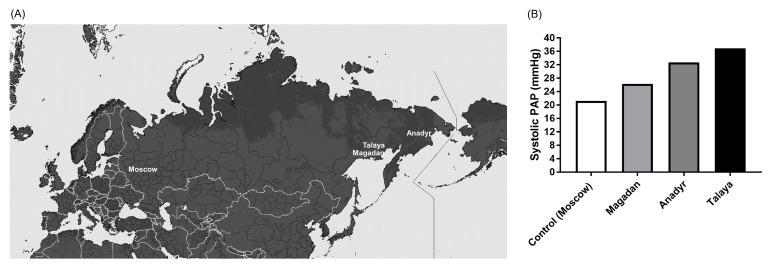
**Increased pulmonary arterial pressure in people living in northeast regions of Russia.** (**A**) Geographic locations of Moscow and northeast Russian human settlements (Magadan, Talaya and Anadyr) characterized by extreme cold are presented. The map is taken from www.maps-for-free.com (accessed on 20 January 2023) and further modified. (**B**) The values of systolic pulmonary arterial pressure (PAP) measured in people living in Moscow (control) and northeast regions of Russia are given. The data derived from one of the studies are taken from the following reference [1].

**Figure 3 life-14-00875-f003:**
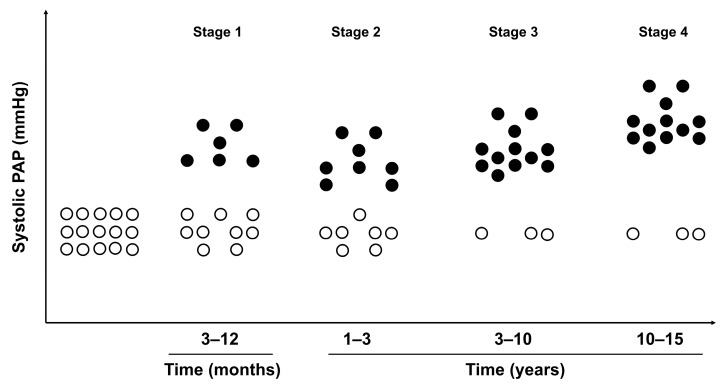
**Time-dependent stages in the development of Northern pulmonary hypertension.** Four stages of the development of Northern pulmonary hypertension are schematically presented by means of the changes in the values of systolic pulmonary arterial pressure (PAP). The circles symbolically represent the human population that spent the given time periods in extremely cold environments. Black-colored circles represent the portion of the population with increased values of the systolic PAP.

**Figure 4 life-14-00875-f004:**
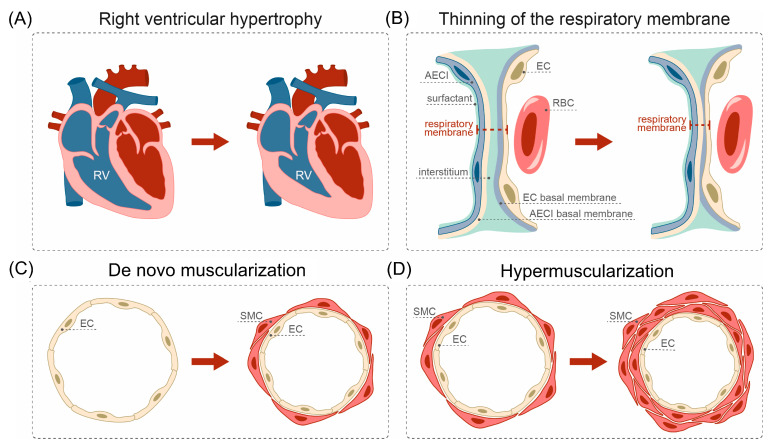
Morphological and histological changes of the cardiovascular and pulmonary systems in people living in extremely cold regions. Some of the alterations of the cardiovascular and pulmonary systems due to extreme and long-term cold exposure include (**A**) right ventricular (RV) hypertrophy, (**B**) “thinning” of the respiratory membrane in the lungs, (**C**) de novo muscularization and (**D**) hyper-muscularization of the pulmonary vessels. Legend: AECI: type I alveolar epithelial cells, EC: endothelial cells, RBC: red blood cells, SMC: smooth muscle cells.

**Figure 5 life-14-00875-f005:**
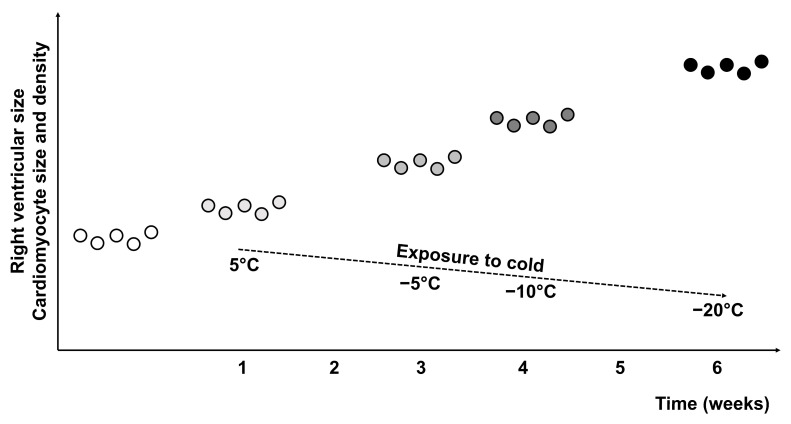
**Development of right ventricular hypertrophy in rats exposed to gradual decrease in temperature.** The development and time-dependent progression of right ventricular hypertrophy is schematically presented by means of the changes in right ventricular size and cardiomyocyte size and density in rats exposed to cold environments. The circles symbolically represent the rat population gradually exposed to cold from 5 °C to −20 °C.

## Data Availability

The data are available from corresponding authors upon reasonable request.

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
