# Peer review of "Northern Pulmonary Hypertension: A Forgotten Kind of Pulmonary Circulation Pathology"

_life, 2024, doi:10.3390/life14070875_

Round 1

Reviewer 1 Report

Comments and Suggestions for Authors

Kosanovic et al present an interesting review of data on cold exposure and pulmonary vascular responses that have not to date been accessible to the non-Russian-speaking community. There is great interest in the idea of environmental exposures that can alter risk or severity of PAH. This notion of cold exposure is also relevant to climate change. 

The review is well-written as well. 

I have some suggested revisions to improve the manuscript: 

1. The studies that are described in both humans and rodents would be more useful if the authors could offer more technical description of the experimental setup, methods of quantifying pulmonary vascular hemodynamics (catheterization, echocardiography, etc.). Currently, the descriptions are too vague to be of great use to the scientific community for replication of the results. 

2. Can the authors comment on any data about the reversibility or irreversibility of the pulmonary vascular and RV changes in cold-exposed individuals? 

3. Is there any histology from animals or humans that can be shown in figures after cold exposure to document pulmonary vascular remodeling? 

4. Can the authors comment on data regarding the combination of altitude (hypoxia) and cold and whether there is a more pronounced effect on severity of PH? 

5. Can the authors comment on whether the populations of Russian cohorts studied in these papers were of the same racial/ethnic groups? Were there certain groups more prone to disease manifestations?

6. Was there a female vs. male predominance of either frequency or severity of disease after cold exposure?

Author Response

Reviewer 1

Q1. Kosanovic et al present an interesting review of data on cold exposure and pulmonary vascular responses that have not to date been accessible to the non-Russian-speaking community. There is great interest in the idea of environmental exposures that can alter risk or severity of PAH. This notion of cold exposure is also relevant to climate change. 

The review is well-written as well. 

R1. We thank to the reviewer for the nice overview of our study and for the support of our manuscript.

Q2. I have some suggested revisions to improve the manuscript: 

  1. The studies that are described in both humans and rodents would be more useful if the authors could offer more technical description of the experimental setup, methods of quantifying pulmonary vascular hemodynamics (catheterization, echocardiography, etc.). Currently, the descriptions are too vague to be of great use to the scientific community for replication of the results. 

R2. We thank to the reviewer for this important comment. The studies analyzed in our manuscript were performed a several decades ago, before the introduction of the echocardiography as the main non-invasive method used in today´s research for quantification of pulmonary vascular hemodynamics. Since the geographic locations investigated in the mentioned studies are remote and characterized by harsh conditions, the catheterization was not feasible. However, in agreement with the reviewer, we have provided the details about the method which was used in the analyzed studies for the quantification of the pulmonary vascular hemodynamics in the revised manuscript, as follows (marked in red, page: 7): “The measurement of sPAP was carried out by the method of Burstin (external graphic recordings) with modification of Dembo and Shapkayts [14]. The Burstin method was based upon measuring the isovolumetric right ventricular relaxation time by means of the phonocardiogram and jugular phlebogram. The accuracy of this method of indirect determination of sPAP was confirmed by comparisons with direct methods and therefore found application in clinical practice [15, 16].” With regard to the study in rodents, the focus was on research of right ventricular hypertrophy. Therefore, the described study investigated the development of right ventricular hypertrophy in rats exposed to extreme cold. In agreement with the reviewer, we have added the description of the experimental setup in the revised manuscript, as follows (marked in red, pages: 13-14): “After the sacrifice, the hearts of rats were taken out and dissected. The right ventricle was separated from the left ventricle and septum and weighted on analytical scales. The measurement of right ventricular hypertrophy was performed via the calculation of the ventricular index. The ventricular index was calculated as the ratio of the mass of the right ventricle with the corresponding part of the interventricular septum to the same mass of the left ventricle.”

Q3. 2. Can the authors comment on any data about the reversibility or irreversibility of the pulmonary vascular and RV changes in cold-exposed individuals? 

R3. We thank the reviewer for raising up this important issue. Up to date, there are no data in the existing literature with regard to the question whether the alterations in the pulmonary vasculature and right ventricle are reversible or not upon the return of individuals to the environment not characterized by the extreme cold. Therefore, the future studies are needed to resolve this important issue. We have addressed this point in the revised manuscript version, as follows (marked in red, page: 18): “Up to date, there are no data in the available literature to indicate whether the alterations of the pulmonary vasculature or right ventricle are reversible or not upon the return of the individuals to the environment not characterized by the extreme cold. Therefore, the future studies should focus to resolve this important scientific and clinical issue.”

Q4. 3. Is there any histology from animals or humans that can be shown in figures after cold exposure to document pulmonary vascular remodeling? 

R4. Unfortunately, we do not possess any original histological data, but we have extensively analyzed the existing literature sources with regard to the pulmonary vascular remodeling. Importantly, we have described in details the histological characteristics of the pulmonary vascular remodeling, which appears due to exposure to the extremely cold environment, as follows (marked in red, page: 11): “According to Milovanov [1], the adaptation of pulmonary circulation occurs in the following sequence: enlargement of alveolar capillaries with increase in the functional zone of respiratory membrane leads to hypervolemic rearrangement of resistance vessels with subsequent hypermuscularization of large branches of the pulmonary artery and increase in the capacity of distributing arteries, which ultimately contributes to hypertrophy of the right ventricle of the heart. The changes described above are characteristic of adaptation and stress [20]. Disorganization of the elastic framework of the trunk and pulmonary arteries with development of elastolysis, widening of gaps by elastic membranes with their partial fragmentation and disorderly twisting, straightening of elastic membranes, formation of a muscular layer in the upper third of the media, multiplication of the internal elastic membrane as a component of the intima, formation of circular smooth muscles in the middle third of the media sometimes with hyperelastosis of the intima, were revealed in the visiting residents of the Far Asian North.” In addition (page: 26), we have shown the schematic representation of the histological changes of the pulmonary vasculature due to cold exposure in the Figure 4 (C and D).

Figure 4. Morphological and histological changes of the cardiovascular and pulmonary systems in people living in the extremely cold regions. Some of the alterations of the cardiovascular and pulmonary systems due to extreme and long-term cold exposure include: A) right ventricular (RV) hypertrophy, B) “thinning” of the respiratory membrane in the lungs, C) de novo muscularization and D) hyper-muscularization of the pulmonary vessels. Legend: AECI: type I alveolar epithelial cells, EC: endothelial cells, RBC: red blood cells, SMC: smooth muscle cells.

Q5. 4. Can the authors comment on data regarding the combination of altitude (hypoxia) and cold and whether there is a more pronounced effect on severity of PH? 

R5. We are grateful to the reviewer for raising up this relevant question. In full agreement with the reviewer, we have included the existing data regarding the combination of high altitude and cold in the revised manuscript, as follows (marked in red, pages: 15-16): “Early studies in cattle suggested the contribution of cold to the high altitude hypoxia in affecting the pulmonary circulation of those animals [28, 29]. Recently, we have investigated the short and long-term effects of cold exposure in Kyrgyz highlanders [25-27]. With regard to the acute effects of cold, the data from our study demonstrated that short-term cold exposure resulted in increase of pulmonary artery pressure in Kyrgyz high altitude dwellers [25]. In the following study, we have found an increase of pulmonary artery pressure (estimated by the tricuspid regurgitant systolic pressure gradient), although not statistically significant, in Kyrgyz lowlanders during the cold season [26]. Importantly, the effects of cold were even more prominent in Kyrgyz highlanders and the data showed a significant elevation of pulmonary artery pressure during the cold season, as compared to the warm season [26]. Therefore, the existing results clearly indicated the existence of the interaction between the cold and high altitude hypoxic milieu.”

Q6. 5. Can the authors comment on whether the populations of Russian cohorts studied in these papers were of the same racial/ethnic groups? Were there certain groups more prone to disease manifestations?

R6. Different studies investigated people of various racial/ethnic origin, including the native inhabitants as well as the newcomers (visiting Northerners). In our manuscript, we have identified that the residents and newcomers/immigrants examined in extremely cold regions (Arkhangelsk, Magadan, Yakutia) and control population (Moscow, Blagoveshchensk), were either of Russian origin or natives, such as Nenets, Chukchi and Yakuts (marked in red, page: 7). With regard to the question whether some groups of different origin are more prone to the disease manifestations, the data are not uniform and not fully conclusive, but there are some interesting findings, that we have described in the manuscript. For example, in the natives, the variation of sPAP was from 19 to 64 mmHg. In the natives of Magadan region, the increase of pressure above 30 mmHg was observed in 25-80% of observations depending on the area of habitat. Interestingly, it was shown that the most pronounced increase in sPAP was found in the inhabitants of the Polar region, especially in those who came from temperate latitudes (marked in red, page: 8). Furthermore, the energy consumption of the right ventricle to promote 1 liter of blood in the visiting Northerners increased in 70% of the examined, while in the natives it was observed only in 45% (marked in red, page: 8). Finally, the higher rates of gas diffusion in the lungs of the indigenous inhabitants of the North were found compared to the newcomers due to more advanced ultrastructural adaptation of respiratory membrane (marked in red, page: 10). Despite these findings, the future comprehensive studies are still needed to profoundly reveal the existence of the favorable adaptations of indigenous inhabitants to the living in the cold regions.

Q7. 6. Was there a female vs. male predominance of either frequency or severity of disease after cold exposure?

R7. This is indeed an important question and one of the studies indicated that the increase in systolic pulmonary artery pressure in males and females was comparable. However, this phenomenon was not systematically investigated and future comprehensive studies are warranted. In agreement with the reviewer, we have included this point in the revised manuscript, as follows (marked in red, pages: 7-8): “It was shown that the increase in sPAP in males and females was comparable. However, the potential gender differences with regard to frequency or disease severity after cold exposure were not profoundly investigated and the future studies should address this important issue”.

Reviewer 2 Report

Comments and Suggestions for Authors

The main purpose of the paper titled "Northern pulmonary hypertension: a forgotten kind of the pulmonary circulation pathology" is to summarize and present the existing knowledge about Northern pulmonary hypertension (NPH) in English, as the majority of the literature on this subject is available only in Russian. The authors aim to detail the pathological characteristics of NPH, a condition that occurs in individuals living in extremely cold regions, and to highlight the need for further studies to uncover the molecular mechanisms behind this condition. This review seeks to bridge the gap in understanding by compiling and translating key findings from Russian studies, making them accessible to the global scientific community. 

Possible improvements for the paper:

Expand on Molecular Mechanisms: While the paper mentions that the profound molecular mechanisms of NPH are yet to be revealed, a more detailed discussion on current hypotheses or preliminary findings would add depth to the paper. Including insights from molecular studies could provide a clearer direction for future research

Include More Comparative StudiesComparing NPH with other types of pulmonary hypertension in different environmental contexts could provide a broader understanding of how cold environments uniquely impact pulmonary vasculature. For example, this recent study 10.5603/CJ.a2021.0159 by Rossi et al can provide a broader perspective on how various environmental and pathological conditions, including extreme cold and viral infections, contribute to PH development.

Enhance Visual Aids and Data Presentation: The paper could benefit from more detailed figures and tables that summarize key data points, such as the differences in pulmonary arterial pressures and right ventricular hypertrophy stages. Clear visual aids can help readers quickly grasp complex information. 

Discuss Broader Implications: Including a section on the broader implications of the findings, such as potential impacts on public health strategies, clinical practices in cold regions, or global health policies, could enhance the relevance of the paper. This discussion could also cover how these findings might inform the management of populations living in other extreme environments.

Address Potential Confounding Factors: A more thorough discussion of potential confounding factors, such as genetic predispositions, lifestyle differences, and other environmental factors, could strengthen the conclusions drawn in the study. Acknowledging and addressing these variables would improve the robustness of the findings. 

Author Response

Reviewer 2

Q1. The main purpose of the paper titled "Northern pulmonary hypertension: a forgotten kind of the pulmonary circulation pathology" is to summarize and present the existing knowledge about Northern pulmonary hypertension (NPH) in English, as the majority of the literature on this subject is available only in Russian. The authors aim to detail the pathological characteristics of NPH, a condition that occurs in individuals living in extremely cold regions, and to highlight the need for further studies to uncover the molecular mechanisms behind this condition. This review seeks to bridge the gap in understanding by compiling and translating key findings from Russian studies, making them accessible to the global scientific community. 

R1. We thank to the reviewer for the nice overview of our manuscript.

Q2. Possible improvements for the paper:

Expand on Molecular Mechanisms: While the paper mentions that the profound molecular mechanisms of NPH are yet to be revealed, a more detailed discussion on current hypotheses or preliminary findings would add depth to the paper. Including insights from molecular studies could provide a clearer direction for future research

R2. We agree with the reviewer that this is an important point. Therefore, we have included some existing (and still very limited) data about the potential molecular mechanisms underlying the development of cold-induced pulmonary hypertension. The data are based on animal rat model, and may provide a direction for future research in the context of human disease. We have included this issue in the revised manuscript version, as follows (marked in red, pages: 15-16): “Although the precise molecular mechanisms underlying the pathology of cold-induced pulmonary hypertension in humans are not known, there are some important data derived from the experimental rat model [33, 34]. Two studies revealed that in the rats exposed to cold there is an upregulation of several signals, including the augmented expression of phosphodiesterase-1C, interleukin-6 and tumor necrosis factor-alpha, as well as increased macrophage infiltration, superoxide production and nicotinamide adenine dinucleotide phosphate oxidase activity [33, 34]. Interestingly, these molecular signals and events associated with altered phosphodiesterase and abnormal inflammation and oxidative stress, are already well-known as relevant players in the pathology of other forms of pulmonary hypertension. This data are useful to provide a future research direction in the context of human disease.”

Q3. Include More Comparative Studies: Comparing NPH with other types of pulmonary hypertension in different environmental contexts could provide a broader understanding of how cold environments uniquely impact pulmonary vasculature. For example, this recent study 10.5603/CJ.a2021.0159 by Rossi et al can provide a broader perspective on how various environmental and pathological conditions, including extreme cold and viral infections, contribute to PH development.

R3. We agree with the reviewer that the comparison of cold-induced pulmonary hypertension with other type of pulmonary hypertension in different environmental context may provide a broader perspective in understanding of the pulmonary vascular pathology in various environmental conditions. Therefore, in the revised manuscript we have described the interaction between cold and high altitude hypoxia in the pathology of pulmonary vasculature, as follows (marked in red, pages: 15-16): “Early studies in cattle suggested the contribution of cold to the high altitude hypoxia in affecting the pulmonary circulation of those animals [28, 29]. Recently, we have investigated the short and long-term effects of cold exposure in Kyrgyz highlanders [25-27]. With regard to the acute effects of cold, the data from our study demonstrated that short-term cold exposure resulted in increase of pulmonary artery pressure in Kyrgyz high altitude dwellers [25]. In the following study, we have found an increase of pulmonary artery pressure (estimated by the tricuspid regurgitant systolic pressure gradient), although not statistically significant, in Kyrgyz lowlanders during the cold season [26]. Importantly, the effects of cold were even more prominent in Kyrgyz highlanders and the data showed a significant elevation of pulmonary artery pressure during the cold season, as compared to the warm season [26]. Therefore, the existing results clearly indicated the existence of the interaction between the cold and high altitude hypoxic milieu.”

Q4. Enhance Visual Aids and Data Presentation: The paper could benefit from more detailed figures and tables that summarize key data points, such as the differences in pulmonary arterial pressures and right ventricular hypertrophy stages. Clear visual aids can help readers quickly grasp complex information. 

R4. We fully agree with the reviewer´s suggestion that the visual presentation can help the readers in easier understanding of the complex information. Therefore, we have included 2 figures (Figure 3 and Figure 5) that clearly show the changes in pulmonary artery pressure and right ventricular hypertrophy stages during the exposure to cold (Figure 3 and Figure 5, marked in red, pages: 25, 26, 27).

Figure 3. Time-dependent stages in the development of the Northern pulmonary hypertension. Four stages of the development of the Northern pulmonary hypertension are schematically presented by means of the changes in the values of systolic pulmonary arterial pressure (PAP). The circles symbolically represent the human population that spent the given time periods in the extremely cold environments. Black colored circles represent the portion of the population with increased values of the systolic PAP.

Figure 5. Development of right ventricular hypertrophy in rats exposed to gradual decrease in temperature. The development and time-dependent progression of right ventricular hypertrophy is schematically presented by means of the changes in right ventricular size and cardiomyocyte size and density in rats exposed to cold environment. The circles symbolically represent the rat population gradually exposed to cold from 5°C until -20°C.

Q5. Discuss Broader Implications: Including a section on the broader implications of the findings, such as potential impacts on public health strategies, clinical practices in cold regions, or global health policies, could enhance the relevance of the paper. This discussion could also cover how these findings might inform the management of populations living in other extreme environments.

R5. We thank to the reviewer for raising up these important issues. In full agreement with the reviewer, we have added in the revised manuscript the new paragraph, which covered the relevant topics, such as clinical practices, public and global health issues and the lifestyle, as follows (7. Further implications: lifestyle, clinical practices, public and global health issues, marked in red, pages: 16-17): “The cold as an important ecological factor is present not only in the geographic locations that are the main focus of this article (North, Northeast and Polar regions), but also in high altitude environments and during the winter time in moderate climates. Therefore, the effects of cold exposure on pulmonary vasculature are of general and global interest. As already mentioned the human populations living in these environments are often isolated from each other and are significantly remote from the “mainland” [3]. The human settlements of the North and Polar regions are not characterized by the existence of the fully contemporary civilization facilities, including the good roads, stable electric supply or internet connections. Importantly, these locations possess the limited medical resources and equipment and do not have the modern clinics. Therefore, the medical care and clinical practice are very challenging as indicated by these relevant public health issues. However, the people living in extremely cold environments, particularly the native inhabitants, developed several behavioral and cultural “adaptations” and established the unique lifestyle with regard to the proper clothing against the dissipation of body heat, warm “shelters” and a special diet [35].

Furthermore, the literature suggested that the knowledge about the impact of the cold on the pulmonary circulation is of the global interest. For example, the study from Israel demonstrated that the values of pulmonary artery pressure were higher during the colder seasons in ambulatory patients who were assessed by echocardiography [36]. Finally, it was shown that there were effects of low temperature on the pathogenesis of other forms of pulmonary hypertension, not associated with extreme environments [37]. Briefly, the cold exposure worsen the development of the disease in the experimental monocrotaline model of pulmonary arterial hypertension.”

Q6. Address Potential Confounding Factors: A more thorough discussion of potential confounding factors, such as genetic predispositions, lifestyle differences, and other environmental factors, could strengthen the conclusions drawn in the study. Acknowledging and addressing these variables would improve the robustness of the findings. 

R6. With regard to the question of other environmental factors, we have added the description of the interaction between cold and high altitude hypoxia in the revised manuscript, as follows (marked in red, pages: 15-16): “Early studies in cattle suggested the contribution of cold to the high altitude hypoxia in affecting the pulmonary circulation of those animals [28, 29]. Recently, we have investigated the short and long-term effects of cold exposure in Kyrgyz highlanders [25-27]. With regard to the acute effects of cold, the data from our study demonstrated that short-term cold exposure resulted in increase of pulmonary artery pressure in Kyrgyz high altitude dwellers [25]. In the following study, we have found an increase of pulmonary artery pressure (estimated by the tricuspid regurgitant systolic pressure gradient), although not statistically significant, in Kyrgyz lowlanders during the cold season [26]. Importantly, the effects of cold were even more prominent in Kyrgyz highlanders and the data showed a significant elevation of pulmonary artery pressure during the cold season, as compared to the warm season [26]. Therefore, the existing results clearly indicated the existence of the interaction between the cold and high altitude hypoxic milieu.”

The issue of the lifestyle has been covered together with other relevant aspects in the reply to the previous question of the reviewer, in the separate chapter of the revised manuscript (marked in red, pages: 16-17): “7. Further implications: lifestyle, clinical practices, public and global health issues.”

To the best of our knowledge, there are no data with regard to the genetic predispositions and development of the cold-induced pulmonary hypertension in humans and this issue remains to be investigated in the future. However, there are findings that suggest the potential existence of some inherited “favorable adaptations” in the natives living in extremely cold environments. For example, the energy consumption of the right ventricle to promote 1 liter of blood in the visiting Northerners increased in 70% of the examined, while in the natives it was observed only in 45% (marked in red, page: 8). In addition, the higher rates of gas diffusion in the lungs of the indigenous inhabitants of the North were found compared to the newcomers due to more advanced ultrastructural adaptation of respiratory membrane (marked in red, page: 10). Despite these findings, the future comprehensive studies are still needed to profoundly reveal the existence of the favorable adaptations of indigenous inhabitants to the living in the cold regions.

Round 2

Reviewer 1 Report

Comments and Suggestions for Authors

The authors have addressed most of my concerns. It is an interesting concept and should be a point of important discussion. However, because of the sometimes questionable nature of the data cited and inability of the reader to discern the accuracy and rigor of the experiments described from the cited Russian papers, the authors should describe the limitations of interpreting the data more directly in the discussion section (add a "Limitations" paragraph that specifically addresses these points). 

Author Response

Reviewer 1

Q1. The authors have addressed most of my concerns. It is an interesting concept and should be a point of important discussion. However, because of the sometimes questionable nature of the data cited and inability of the reader to discern the accuracy and rigor of the experiments described from the cited Russian papers, the authors should describe the limitations of interpreting the data more directly in the discussion section (add a "Limitations" paragraph that specifically addresses these points). 

R1. We thank to the reviewer for this important comment. In agreement, we have included the “Limitations” paragraph in the revised manuscript, as follows (Limitations, marked in red, pages: 17-18): “We would like to mention that there are certain limitations of our review article, which should be kept in mind while interpreting the data from the existing literature. First, many relevant studies described in this paper were performed a couple of decades ago and some of the methods, for example the assessment of the pulmonary hemodynamics, were performed using the old technical approaches. Therefore, the new studies using the contemporary and more accurate techniques should be applied in the future to validate and prove the old findings. In addition, due to the fact that many literature sources are published several decades ago and written only in Russian language, it is not easy for the readers from other countries to assess the original papers and to discern the accuracy and rigor of the experiments. However, we hope that our paper will intrigue and inspire the researchers in the field to confirm the old findings and provide the new data with regard to this important medical phenomenon, such as cold-induced pulmonary hypertension.”
